# Leveraging Expert Demonstration Features for Deep Reinforcement Learning in Floor Cleaning Robot Navigation

**DOI:** 10.3390/s22207750

**Published:** 2022-10-12

**Authors:** Reinis Cimurs, Emmanuel Alejandro Merchán-Cruz

**Affiliations:** 1SIA Robotic Solutions, LV-1039 Riga, Latvia; 2Transport and Telecommunication Institute, Engineering Faculty, LV-1019 Riga, Latvia

**Keywords:** Deep Reinforcement Learning, mobile robot navigation, semi-supervised learning, autonomous cleaning robots

## Abstract

In this paper, a Deep Reinforcement Learning (DRL)-based approach for learning mobile cleaning robot navigation commands that leverage experience from expert demonstrations is presented. First, expert demonstrations of robot motion trajectories in simulation in the cleaning robot domain are collected. The relevant motion features with regard to the distance to obstacles and the heading difference towards the navigation goal are extracted. Each feature weight is optimized with respect to the collected data, and the obtained values are assumed as representing the optimal motion of the expert navigation. A reward function is created based on the feature values to train a policy with semi-supervised DRL, where an immediate reward is calculated based on the closeness to the expert navigation. The presented results show the viability of this approach with regard to robot navigation as well as the reduced training time.

## 1. Introduction

Over the last couple of decades, robotic technologies have proliferated in various fields with a high rate of repetitive and manual labor-intensive tasks. Highly repetitive tasks could be described by a set of commands and automated without requiring complex decision-making mechanisms. Expert demonstrations were enough to encode the robot states and reactions to them in safe and static environments. With recent advancements in Deep Learning (DL), machine learning technologies are more capable of being applied in highly dynamic environments and solving complex tasks in ambiguous settings. Such functions as floor vacuuming, lawn mowing, and even autonomous driving have been delegated to autonomous systems with differing levels of artificial intelligence [1,2]. Their necessity for human-like behavior is highly dependent on their domain’s requirements, and the performance’s optimality is not always required. In navigation tasks for floor cleaning robots, it is not only important to perform the cleaning without wasting resources, but also to do it in a manner that resembles the experience of a qualified operator. It should be reasonably understandable to non-experts in order to build confidence in the robot’s observable skills, as cleaning environments are often shared with workers or clients of the respective establishment. A Behavioral Cloning (BC) approach is often used to embed the robot with human-like navigation. Recorded human trajectories are used to learn how to replicate them when encountering the same situation. However, BC has reported issues when dealing with generalization [3]. For generalization, it requires a specific setting or a large amount of unbiased expert data. The expert demonstrations alone are not enough to gain all the required knowledge in some domains where the state-space is so large that it is essentially impossible to collect all the relevant experiences in order to learn a generalizable policy. Deep Reinforcement Learning (DRL) is a popular method to overcome this bottleneck. DRL learns in its given environment through trial and error, optimizing its policy by evaluating its reactions to states through the returned reward. A state representation is input into the neural network, and the resulting action is obtained. This action is then performed in the environment, and a reward is calculated based on the new state. Then the state–action pair can be optimized based on the returned reward. However, engineering this reward in DRL is a daunting task, especially when a particular behavior is desired. It is difficult to predict what behavior will result from a hand-engineered function and whether it will be acceptable for other human agents in the environment. Therefore, we propose to implement the idea, borrowed from BC, to use the expert demonstrations and assume them as a guideline for training a neural network for cleaning robot navigation by using DRL. We propose to collect a dataset of expert demonstrations, define characteristic features of motion in the floor cleaning domain, and calculate a reward function based on them. Then use this reward to train a neural network through trial and error, without needing extensive expert data, while maintaining their characteristic behavior. During the training, expert demonstrations can be used as training samples to expose the neural network to optimal navigation, thus speeding up the learning process. Our contributions in this paper can be itemized as follows:Introduce expert motion features in the floor cleaning robot domain.Develop a reward calculation method from expert motion features.Implement an expert replay buffer in the DRL pipeline and introduce a combined batch creation with an experience replay buffer for optimized training.

The remainder of this paper is organized as follows. In Section 2 related works are reviewed. The method to obtain an expert feature-based reward is described in Section 3. The proposed neural network architecture is discussed in Section 4 and its training setup presented in Section 5. Experimental results are given in Section 6 and conclusions given in Section 7.

## 2. Related Work

The relevant related works to the proposed cleaning robot navigation commands by leveraging expert demonstrations with DL can be separated into two parts. First, the proposed method aims to perform learned motion in continuous action space by employing a neural network in order to obtain smooth and immediate motion output to incoming environment description. Therefore, the relevant research in autonomous robot motion control through DL is discussed. Second, the learned motion policy introduces expert experience into the training process while maintaining the exploration strategies inherent to RL, so methods of introducing expert demonstrations into neural network-based approaches are reviewed.

### 2.1. Neural Network-based Navigation

With the advancement of Deep Learning, there has been a significant amount of work focusing on creating autonomous navigation systems based on neural networks [4,5]. While many of these works focus on high-level planning and leave the execution of control and motion to low-level controllers [6,7], various methods of direct action execution from sensor inputs have also been considered [8]. A popular DL-enabled motion policy training method is performed with neural networks based on the Q-learning approach [9,10,11,12]. A set of possible action outputs is given to a neural network and a Q-value estimation for each action is learned, conditioned on the input state. This method learns to evaluate the possible future reward for all the given actions at each timestep and selects one of them for execution based on set criteria. In [13], a robot learns navigation behavior directly from the depth images of the environment and is capable of selecting one of five actions to avoid collisions while moving in an unknown environment. However, such approaches require a specified discrete set of actions. Since these algorithms cannot output actions in continuous action space, they are unsuitable for mimicking expert behavior in the floor cleaning robot domain, where human inputs are recorded and executed as analogous signals.

Deep Deterministic Policy Gradient-based (DDPG) architectures are often used to tackle mobile robot navigation tasks in continuous action space. DDPG is an actor–critic-based model where the actor network outputs motion controls as real scalars, and the critic evaluates the state–action pair. In [14], sparse laser inputs are used to train a DDPG-based neural network to navigate in a simulated environment. In [15], a Soft Actor–Critic (SAC) algorithm is used instead of DDPG to achieve similar results to those in [14]. However, the simplified state representation has difficulty navigating around objects of complex shapes, as shown in [16], where the DDPG network is extended to include temporal image embedding through a Convolutional Neural Network (CNN). However, the inclusion of the CNN drastically increases the number of network parameters, making it difficult to transfer it to real robot devices. A negative aspect of the DDPG algorithm is its tendency to overestimate the Q-value, thus making the learning unstable [17].

To counter this issue, a Twin Delayed Deep Deterministic Policy Gradient (TD3) architecture uses two critic networks instead of one and chooses the values of the critic that provides the lower values, thus limiting the overestimation [18]. TD3-based algorithms have been used to successfully train stable neural network-based motion policies [19,20]. In the mobile robot domain, the authors in [21] develop a TD3-based approach for combining path planning and motion execution and report outperforming DDPG and SAC approaches. In [22], a TD3 neural network is used as a backbone for a goal-oriented exploration algorithm in unknown environments. Both approaches show the capacity of TD3 to carry out local robot motion with a high success rate, which can be used in real robot motion scenarios. Therefore, we have taken this information into account and selected a TD3-based architecture for implementing learning motion policies in continuous action space with a reward function derived from expert motion features.

### 2.2. Expert Feature Extraction

Expert knowledge exploitation in mobile robots has been an active research area for decades [23]. From expert systems, where fuzzy control is derived from recorded expert motions [24], to BC-based methods, where a policy is learned directly from expert demonstrations [25], prior knowledge helps to develop motion policies with human-like behavior. BC sets the policy learning as a supervised learning task that trains a model to predict an expert action from environmental input and has been widely used in autonomous driving scenarios [26,27]. The role of an expert in the autonomous car driving domain is well understood, and the backing of large corporations has allowed to collect and freely distribute various driving datasets. Generally, this is not the case for mobile robots, which are mostly specialized devices performing specific tasks and, as such, do not have freely available expert motion datasets. Therefore, authors have to rely on manually collected data and deal with the associated issues. Learning from manually collected human demonstrations based on goal point and laser data is proposed in [28]. In [29] the authors train a neural network to navigate a personal mobility device in a small area based on RGB images, overcoming the issues related to a small and noisy dataset. Authors in [30] develop a lifelong learning network that improves its policy from an initial policy based on the Dynamic Window Approach (DWA) planner. However, these approaches either require extensive data for the environments that they are trained in or do not generalize well in unknown spaces.

Another way of embedding the motion policy with human-like behavior is to extract the reward function from expert actions and use it to evaluate the learned policy during the training. This method is referred to as Inverse Reinforcement Learning (IRL) and is widely used in trajectory prediction and action selection tasks [31]. In the mobile robot domain, IRL has been successfully applied in [32] to predict and execute trajectories observed from human motion. Sample efficient IRL for robot navigation from human motion is proposed in [33]. However, the ego agent is conditioned only on other agents in the simulation and not on the surroundings. Navigation based on environment features is introduced in [34], but the learned actions are considered in discrete action space. In general, IRL methods suffer from challenges in tasks, such as proper feature selection, sensibility to features, difficulty in posing the IRL problem and lack of appropriate data to query [35]. To solve the data querying problem, RL methods are still required. Methods without using IRL, but still infusing human supervision in RL setting, are proposed in [36,37]. Here, instead of obtaining the expert knowledge beforehand, a human-in-the-loop approach is used to query an expert for advice on action and reward. While beneficial to the overall progress of policy learning, this requires active interaction of human experts during the online training, which can be cumbersome and time-consuming. In our implementation, we propose to combine the BC-based approach of evaluating the performance against the designed expert motion features in the cleaning robot domain, which are expressed as a reward function, thus allowing to use DRL for learning a motion policy in unknown environments without the need to learn the reward function through IRL.

## 3. Expert Feature-Based Reward Engineering

Neural network weight optimization in DRL is facilitated by the selected reward function. In guided robot motion tasks, the main reward selection criteria are rather clear and well documented—if the goal is reached, the environment returns a high positive reward, but if a failure state is reached, the environment returns a large negative reward. However, training a policy based on just these two reward criteria can be very difficult, especially if the returned rewards are sparse. The neural network might not encounter any of these states during the exploration, thus limiting the learning capabilities from such trajectories. Additionally, this excludes embedding the motion with any kind of behavior, as the policy will optimize only reaching the goal and not the motion itself. To rectify this issue, an immediate reward function is often used. Each individual motion step is evaluated, and a reward is attributed to it. Through this immediate reward, it is not only possible to guide the policy to the goal, but also embed a desired behavior of the motion. In the case of robot motion for floor cleaning tasks, prior knowledge can be used to engineer the features of the immediate reward function. For natural motion, the robot should control its linear and angular velocities with respect to obstacles and select a natural turning radius considering its goals. This allows for the use of floor cleaning expert domain knowledge and determines four motion parameters that are selected as the main features for describing the expert trajectories:Linear velocity with respect to distance from obstacles.Angular velocity with respect to distance from obstacles.Linear velocity with respect to orientation to the goal.Angular velocity with respect to orientation to the goal.

The described features allow for detection of speed and turning rate of the expert when faced with obstacles and heading towards a goal and calculate an immediate reward function based on them.

### 3.1. Expert Demonstration Collection

A database of expert demonstrations is collected to calculate the reward function from motion features. It is assumed that the cleaning strategy follows a global plan, delegating target points in the environment and ensuring the coverage of the cleaning surface. Therefore, only the local motion between the target goal points is gathered to learn the motion policy. A human is tasked to navigate in a simulated environment between randomly placed goal points. Navigation is performed by inputting continuous actions using analogous joystick commands, the action is translated into linear and angular velocities and executed in the simulation. Each expert action execution as well as the state of the environment is recorded as the following tuple:(1)$$h_{i}=\left(l_{i},\delta_{i},\theta_{i},v_{i},\omega_{i}\right)$$
where the state history *h* consists of lidar scan data *l*, distance to the goal δ, heading difference to the goal θ, linear velocity *v* and angular velocity ω at the iteration *i*.

### 3.2. Expert Feature Coefficient Calculation

To simplify the expert feature expressions, a linear relation is assumed between the action taken by the expert and the relevant trigger for it—closeness to the obstacle or heading difference to the goal. That allows calculating a single coefficient value for each feature that describes the expert behavior. Working with this assumption, in floor cleaning scenarios, prior domain knowledge is used to conclude that linear velocity decreases and angular velocity increases with the decreased proximity to an obstacle as the robot needs to navigate away from it safely. On the other hand, the linear velocity increases while angular velocity decreases with the decrease in heading difference between the robot and the goal. This allows the formation of an equation to calculate the expert action values, dependent on feature coefficients:(2)$$v=l_{min}\times o_{v}$$
(3)$$\omega =\frac{o_{\omega }}{l_{min}}$$
(4)$$v=\frac{\phi_{v}}{\theta }$$
(5)$$\omega =\theta \times \phi_{\omega }$$
where $l_{min}$ is the minimum lidar reading signifying the distance to the closest obstacle, $o_{v}$ and $o_{\omega }$ are the coefficients for calculating the linear and angular velocity based on the distance to the obstacle, $\phi_{v}$ and $\phi_{\omega }$ are the coefficients for calculating the linear and angular velocity based on the heading angle difference towards the goal, respectively.

The full set of the collected data is used to calculate the values of coefficients that describe the features of expert actions. Reforming Equations (2)–(5) allows calculating the values of $o_{v}$, $o_{\omega }$, $\phi_{v}$ and $\phi_{\omega }$ over all the iterations of collected data:(6)$$o_{v}=\frac{\sum_{i}^{N_{h}}\frac{v_{i}}{l_{min,i}}}{N_{h}}$$
(7)$$o_{\omega }=\frac{\sum_{i}^{N_{h}}\omega_{i}\times l_{min,i}}{N_{h}}$$
(8)$$\phi_{v}=\frac{\sum_{i}^{N_{h}}v_{i}\times \theta_{i}}{N_{h}}$$
(9)$$\phi_{\omega }=\frac{\sum_{i}^{N_{h}}\frac{\omega_{i}}{\theta_{i}}}{N_{h}}$$
where $N_{h}$ is the total number of expert sample iterations.

### 3.3. Reward Engineering

Knowing the expert feature coefficient values allows for calculating the action values for previously unseen states. Reformulating the Equations (2)–(5) as reward signals, it is possible to evaluate every state–action pair for every expert feature as follows:(10)$$r_{v,t}^{o}=\left|v_{t}-l_{min,t}\times o_{v}\right|$$
(11)$$r_{\omega ,t}^{o}=\left|\omega_{t}-\frac{o_{\omega }}{l_{min,t}}\right|$$
(12)$$r_{v,t}^{\phi }=\left|v_{t}-\frac{\phi_{v}}{\theta_{t}}\right|$$
(13)$$r_{\omega ,t}^{\phi }=\left|\omega_{t}-\theta_{t}\times \phi_{\omega }\right|$$
where $r_{v,t}^{o}$, $r_{\omega ,t}^{o}$, $r_{v,t}^{\phi }$ and $r_{\omega ,t}^{\phi }$ at the time step *t* is the linear velocity reward for the obstacle feature, angular velocity reward for the obstacle feature, linear velocity reward for heading difference feature and angular velocity reward for heading difference, respectively. The smaller the reward, the smaller the difference between the taken action and the action calculated based on the expert feature. Following, the immediate state action reward $r_{t}$ at a single timestep then can be expressed as:(14)$$r_{t}=-r_{v,t}^{o}-r_{\omega ,t}^{o}-r_{v,t}^{\phi }-r_{\omega ,t}^{\phi }$$

The function negatively rewards large deviations from actions calculated by the expert features. To allow for positive rewards, Equation (14) is normalized in the range [−1,1]. The use of the immediate reward function allows for training the policy to follow the behavior of the expert, with the possibility of training on samples outside the available trajectories in the dataset.

## 4. Neural Network Architecture

With an engineered immediate reward function, a neural network has a feedback signal from the environment to update its internal weights and optimize the calculated policy. To facilitate the backpropagation, a Twin Delayed Deep Deterministic Policy Gradient-based (TD3) neural network is designed. TD3 is an actor–critic type neural network architecture that employs two critic networks to stabilize the value function estimation while still optimizing the policy of the actor network.

### 4.1. TD3-Based Architecture

The TD3-based architecture consists of one actor and two critic networks that take as inputs the representation of the environmental state and outputs motion commands of the policy. The parameters of actor and critic networks are not shared and are calculated independently. As an input at the timestep *t*, the actor network *A* takes a tuple of environmental states *s* defined by:(15)$$s_{t}^{A}=\left(l_{t},\delta_{t},\theta_{t},v_{t-1},\omega_{t-1}\right)$$

The values in the state are formed into two single dimension vectors. The first vector consists of all the lidar values *l*, and the second vector consists of combined values of δ, θ, $v_{t-1}$ and $\omega_{t-1}$. Both vectors are presented as an input to the actor network and independently passed through an encoder, consisting of a fully connected layer, followed by the Rectified Linear Unit (ReLU) activation function and another fully connected layer. We refer to these sequences of encoding layers as external state encoder and internal state encoder, respectively. Afterward, external and internal state encoder outputs are concatenated and passed through an encoder to obtain the final state encoding. State encoding is passed through two more fully connected layers, each followed by ReLU activation. Another fully connected layer follows, with its output passed through the Hyperbolic Tangent (Tanh) activation function to obtain the results in the range (−1, 1).

The architecture of the critic networks *C* is similar, with its input state also including the output $a_{t}=\left(v_{t}^{A},\omega_{t}^{A}\right)$ that consists of linear and angular velocities calculated by the actor network and its tuple is defined as:(16)$$s_{t}^{C}=\left(l_{t},\delta_{t},\theta_{t},v_{t-1},\omega_{t-1},a_{t}\right)$$

External and internal state encoding is obtained the same way as in the actor network with the subsequent state encoding. $a_{t}$ is encoded with a single fully connected layer. State and action encoding are combined by using the Transformation Fully Connected (TFC) layer method from [22]. ReLU activation is performed on the TFC layer output and then passed through the final fully connected layer to obtain the estimated Q value of the state–action pair.

### 4.2. Leveraging Expert Samples in Experience Replay

During the training, the neural network employs a replay buffer to collect and store a set of samples on which it will be trained. An action is performed with the policy, the relevant information is obtained and saved in the memory in the form of the following tuple:(17)$$H_{t}=\left(s_{t},a_{t},r_{t},s_{t+1},d_{t+1}\right)$$
where *H* is the obtained transition information between states $s_{t}$ and $s_{t+1}$, $d_{t+1}$ indicates whether the state $s_{t+1}$ is terminal. Owing to the attribute that the proposed TD3 network is an off-policy algorithm, it is possible to learn updated policy by using samples collected from older policies. Nonetheless, keeping all the samples in the replay buffer might force an overfit to the sample data and slow down the learning of the network weights, as well as increase the required memory storage. Therefore, only a limited number $N_{H}$ of the most recent samples are kept in the buffer.

During the early stages of the training, the policy is not yet capable of navigating around the environment and collecting varied samples. It is also difficult to collect data where the policy would lead to the goal, as the motions are largely based on random initialization of the neural network weights. However, there exists a large amount of collected expert demonstration data in various settings that not only show the navigation in the environment but also lead it to sparse goals. Moreover, learning and optimizing for this data is desired as it is representative of the desired optimal behavior of the agent. Since the replay buffer *H* only holds $N_{H}$ number of samples, direct infusion of expert data is not possible, as newly collected samples would replace them over time. Therefore, we propose introducing a secondary expert replay buffer *E* of set size $N_{h}$ consisting of all the expert samples. To create the expert replay buffer, each collected sample from (1) is formed in a tuple form of (17). To form the tuple, two consecutive expert iterations $h_{i}$ and $h_{i+1}$ are combined and evaluated by (14) to obtain the state–action pair reward.

The neural network learns on a batch of samples from the replay buffer. To obtain the batch, both replay buffers *H* and *E* are sampled individually, and the selected samples from each buffer are combined. This allows the network to learn directly from the optimal policy data, as well as generalize to situations unseen during the expert demonstrations. The introduction and working mechanism of this approach are visualized in Figure 1.

## 5. Neural Network Training Setup

To train the neural network, described in Section 4, a pipeline is built with the PyTorch framework. Full network parameters are displayed in Table 1 and Table 2 as well as Figure 2. The Robot Operating System (ROS) Noetic version serves as a middleware between the network output and the execution in the simulation. ROS Gazebo is used as a simulator for executing actions on a simulated agent and gathering experience data from the experts.

### 5.1. Reward Function

To create reward signals for the neural network training, the description of the immediate reward $r_{t}$ function is given in Section 3.3. However, termination states also need to be rewarded. Therefore, the following reward function for neural network training is employed:(18)$$r\left(s_{t},a_{t}\right)=\left\{\begin{array}{cc} r_{g} & \;\mathrm{if}\;\delta_{t}<\eta_{\delta }\\ r_{c} & \;\;\;\;\;\mathrm{if}\;l_{min,t}<\eta_{l}\\ r_{t} & \;\mathrm{otherwise}, \end{array}\right.$$
where $\eta_{\delta }$ is the distance threshold to the goal at which it is considered reached, $\eta_{l}$ is the distance threshold to the closest obstacle at which a collision is considered to have occurred.

### 5.2. Expert Demonstration Collection

To calculate the reward function used in training and collect the samples of optimal policy, a human expert is tasked to perform navigation in a simulated environment. For this purpose, an environment is designed in the ROS Gazebo simulator to replicate a similar learning environment to that of a recorded real-life location of Riga Technical University, where cleaning tasks would be carried out in the future, as visualized in Figure 3a. The designed simulated environment is visualized in Figure 3b. The expert is tasked with navigating to randomly placed goals in the environment using an analogous joystick input. The expert inputs, as well as environmental states, are recorded as described in (1). Each action is executed at 0.1-s intervals, which is the time difference between the iterations. After reaching a goal, the full trajectory of iterative motions is saved in a separate JavaScript Object Notation (JSON) format data file and the environmental state, as well as the goal, are randomly reset and new trajectory data collection begins. To perform reward engineering analysis from the expert demonstration files, all JSON files are loaded into memory, and calculation is performed over all the iterations. When reformatting the expert demonstrations to the form described in (17) for expert replay buffer, each JSON file is loaded individually and the final iteration in each file is assumed to reach the goal and is rewarded with the value of $r_{g}$, according to (18). In total, 160 trajectories were collected with 6448 motion samples.

### 5.3. Simulated Training Environment and State Representation

To train the neural network, another virtual training environment is created, filled with various shapes to help generalize the learned policy. Additionally, virtual cube objects are placed in the environment, and they change their location to a new randomly selected coordinate on every completion of an episode to randomize the learning environment. The environment is designed with ROS Gazebo simulator and visualized in Figure 3c. As an agent, a simulated Pioneer 3DX robot device is used, through which the action execution and observation of the state take place. The state is collected through simulated robots’ odometry and a simulated 16-channel lidar mounted on top of it. Random Gaussian noise is added to sensor data to help generalize the learning process. The information obtained from the odometry and lidar is processed and prepared for use as a one-dimensional, vectorized representation of the environment. The robot’s position in the environment is represented with respect to the goal by $\delta_{t}$ and $theta_{t}$, calculated by comparing the global coordinates of the robot and heading from the odometry information to the goal. Lidar information from the simulation is returned as a 3D point cloud with a 180-degree field of view (FOV). First, filtering out of the ground plane is performed. Then, the data is prepared as a 2D representation of the 3D data, where the minimal distance measurement is returned for any discreet angle in the FOV of the lidar as described in:(19)$$l_{\lambda }=min\left(L_{-90+z\times \lambda_{\delta }}^{c}\right),\;\;\;\;\;\;\;\;\;\;\;\forall z=\left\{1,2,...,\frac{180}{\lambda_{\delta }}\right\},\forall c=\left\{1,2,...,n\right\}$$
where λ is the discretized angle value, $\lambda_{\Delta }$ is the discretization size of λ, *c* is the respective lidar channel and *n* is the number of remaining channels after floor filtering. To reduce the input size of the lidar data, the 2D lidar representation is further discretized by performing data bagging. The $l_{\lambda }$ data is sequentially placed in *b* number of bags of equal length and the minimal value of each bag the representative value of the respective bag. This allows for the creation of a single dimension vectorized representation of *b* number of elements from the full lidar data.

### 5.4. Training Parameters

Examples of neural network parameters are shown in Table 1 and Table 2 as well as Figure 2. The actor-network parameters in Table 1 show the inputs and outputs of each sequential layer. Lidar and goal information is encoded separately, combined by concatenation, and further encoded in the state encoder. Afterward, state encoded information is mapped to an action by the action decoder.

The architecture of the critic networks, described in Table 2, is similar to the actor network. External and internal information is encoded with respective encoders and their combined encoding is obtained by the state encoder. A Q-value decoder follows to estimate the value of the state–action pair. State information is given by the state encoder with an additional sequence of linear layers and ReLU activations performed in the Q-value decoder. $Linear_{state}$ maps this information to 600 parameters through another linear layer. In parallel, action from the state–action pair is also mapped to 600 parameters through the $Linear_{action}$ layer. The encoded state and action information is combined in the TFC layer followed by mapping to a single Q-value with the final output linear layer.

Additional parameters describing the selected sparse rewards, calculated immediate reward coefficient values, robot velocity, and other training-dependent values are depicted in Table 3. The seed represents the set value for random weight initialization. Maximal episode length represents the maximal steps before termination of the training episode if no goal was reached or no collision occurred. $H_{batch}$ represents the number of random samples in a batch selected from experience replay buffer *H*. $E_{batch}$ reflects the number of random samples in a batch selected from expert replay buffer *E*. The full batch size for each learning iteration is $H_{batch}+E_{batch}$ number of samples.

## 6. Results

To evaluate the validity of the proposed approach, a series of experiments in a simulated environment were carried out. The proposed method is compared to other popular immediate reward calculation methods implemented in the TD3 and Deep Deterministic Policy Gradient (DDPG) architectures. Additionally, effects with and without expert replay buffer *E* are presented.

### 6.1. Experiments in Simulated Environment

In general BC tasks, the precision of the learned policy can be evaluated by the closeness of the executed trajectories to the expert trajectories. Since the task of the proposed method is to generalize the policy, the exact closeness to expert demonstrations is not expected. To evaluate the closeness of the learned policy to the expert baseline in action space, a direct comparison is made between the action of an expert and a policy. A state representation is obtained from all the samples in the expert motion database, collected in the simulated environment depicted in Figure 3b. All samples are processed with the trained neural network and the obtained action is directly compared to the expert motion. To show the effects of the expert buffer, the proposed method is trained and evaluated with and without it and the elements are referred to as *Ours w/ buffer* and *Ours w/o buffer*, respectively. To evaluate the expert feature coefficient calculation method, the proposed method is compared to handcrafted immediate reward methods based on the reward used in [16], which are designed to evaluate the expert motion as precisely as possible. These methods are based on an immediate reward function:(20)$$r_{t}=v_{t}-\left|\omega_{t}\right|$$
and
(21)$$r_{t}=v_{t}-\left|\omega_{t}\right|-r_{obs}$$
where
(22)$$r_{obs}=\left\{\begin{array}{cc} 1-l_{min} & \;\mathrm{if}\;l_{min}<1\\ 0 & \;\mathrm{otherwise}, \end{array}\right.$$

Here $r_{obs}$ represents the inverse reward for closeness to the obstacle. These handcrafted methods are referred to as *H.C w/o obs.* and *H.C w/ obs.* for (20) and (21), respectively. A comparison is also made with policies trained with immediate rewards presented in [38,39]. They are referred to as $r_{1}$ and $r_{2}$, respectively. The results are presented in Table 4.

From the experiment results, it can be seen that the performed action on each policy slightly differs from the expert action. In general, policies that have been conditioned on expert trajectory data learn to move with higher linear velocity and the increase in rotational velocity is proportional. This can be explained by introducing a goal reward, which incentivizes the policy to arrive at the goal as quickly as possible. Thus, the policy moves faster, but also has to increase the rotational velocity in order to comply with the expert feature reward. The policies that are not conditioned on expert data show different behavior with either increased rotational velocity or even moving slower than the expert, as in the case of $r_{2}$. Therefore, they exhibit different behavior than the experts.

To compare the learned motion policy with the expert behavior over the whole task of arriving at the goal, another test is carried out in the simulated environment shown in Figure 3b, where expert trajectories were collected. This allows for direct comparison to expert trajectories. Each policy is tasked to perform a motion from a selected starting point to a goal point. Each learned policy’s trajectory is recorded and compared to the expert trajectory. Visualized examples of the experiments are shown in Figure 4. Policies are evaluated on multiple key performance indicators such as the difference in the number of steps taken to arrive at the goal with respect to the expert trajectories, the episode reward $r_{ep}$ as calculated by (14), and the linear and angular velocities. The most important indicator is the L2 difference between each point on their trajectory and the closest point in the expert trajectory. This shows the closeness of the executed trajectory by the learned policy to the expert trajectory. In turn, it implies taking similar motions to experts and reacting to the same feature triggers as in the expert trajectory. The results are presented in Table 5.

From the experiments, it can be seen that the proposed approach follows closer to the expert trajectories, although with increased speed and proportionally increased rotational velocity. This also leads the policy to arrive at the goal position in fewer steps. This corresponds to the observations from Table 4. This is more significantly seen in the performance in the model that uses the samples from the expert replay buffer in the training. Thus, the policy with an expert replay buffer takes approximately four steps less to arrive at the goal than the expert trajectory baseline. Similarly, the method without an expert replay buffer arrives at the goal more quickly, in contrast with other approaches that take longer to arrive at the same goal. While handcrafted reward functions perform better than the proposed method on individual action predictions, they do not generalize as well over the whole trajectory. Curiously, the introduction of the obstacle reward in the handcrafted reward function *H.C. w/ obs.* shows a drastic reduction in L2 scores if compared to *H.C. w/o obs.*, which signifies that the obstacles are a significant feature in expert motion decision-making. $r_{1}$ and $r_{2}$ methods expectedly do not perform well in the task as they are not conditioned on the expert features. While they are capable of optimizing towards their own rewards, their executed trajectories differ from a human expert, especially in the $r_{2}$ case. Even though the obtained reward for $r_{1}$ is quite close to the proposed method, the obtained behavior still differs significantly over the whole trajectory This shows the influence that the immediate reward and the motion features play in behavior embedding for autonomous navigation and the necessity to address it if human-like motion is expected.

To qualitatively show the possibility of the proposed method to adapt to previously unseen situations, experiments in additional designed scenarios are shown in Figure 5. The expert trajectories were collected in set scenarios, but not used for neural network training. From the experimental results, it is possible to visually see that the trained neural network, conditioned on expert trajectories, still performs reasonably close to expert motion, even if the policy has not been trained on such examples.

### 6.2. Training Time Comparison

By introducing the expert replay buffer *E* in batch generation for neural network training, it is possible to speed up the learning process. By adding the samples from the expert trajectory, the neural network batch has the possibility to sample a high sparse reward of reaching the goal. This in turn serves as a bootstrapping method, with introducing the goal reward early on in the training process. As a result, the training of TD3 with batch sampling from *E* converges to the maximum Q-value more quickly than the compared methods. The results of this comparison are visualized in Figure 6.

A similar observation can be seen by evaluating the average Q value convergence. The proposed method with an introduced expert replay buffer increases the average Q value in a more stable manner and reaches the saturation point approximately 4 h into the training time. Other methods exhibit a more unstable policy evaluation convergence and reach the saturation point later in the training. This is most likely due to the random nature of arriving at the sparse rewards and not having a significant number of such encounters early in the learning process to train on. Even though the immediate reward function explicitly guides the robot towards the goal, the proposed method without the buffer *Ours w/o buffer* also exhibits unstable learning behavior during the early stages of training, albeit with smaller oscillations than compared methods. This is visualized in Figure 7.

## 7. Conclusions

In this paper, a method of extracting a reward from expert demonstrations on floor cleaning operations and leveraging their experience in neural network training is introduced. As the experiments show, the proposed method successfully learns a generalizable motion policy for mobile robot navigation. It is capable of learning from expert demonstrations and transfers its motion features to a learned behavior through a designed reward function. The neural network motion policy is able to resemble the expert motion also outside the previously seen samples, which is a benefit over BC-based approaches, and without expensive Inverse Reinforcement Learning reward function estimation. Additionally, as the results in Section 6.2 show, Q-value convergence can be achieved faster than with compared methods.

For future works, the it is planned that the current pipeline will be extended to include different types of real industrial floor cleaning mobile robot platforms and perform tests in real-life settings. In order to achieve this, a method of avoiding dynamic obstacles needs to be developed. To extract dynamic motion information, environment states of previous timesteps should be represented in the neural network input either in the state encoder or using Gated Recurrent Units (GRUs), Recurrent Neural Networks (RNNs), Long-Short Term Memory (LSTM), or other methods. The additional history retention should allow to make more informative motion policy decisions, especially when dealing with dynamic obstacles. Additionally, the method in the proposed approach of extracting the expert features assumes a linear relation. While in floor cleaning tasks, where absolute expert behavior precision is not detrimental, it is sufficient, a non-linear relation will also be explored. This might not only provide the benefit of more closely replicating the expert behavior, but also provide better gradients for neural network learning. These are the current active directions that should prove to be beneficial in the next iterations of this research topic.

## Figures and Tables

**Figure 1 sensors-22-07750-f001:**
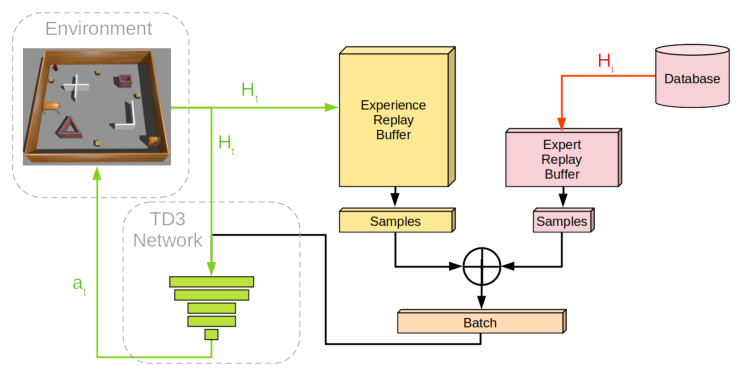
Flowchart of expert experience leveraging in the neural network training process. Red arrows represent actions that happen once during the initialization of neural network learning. Black arrows represent processes happening once every training cycle. Green arrows represent processes taking place during every timestep *t* of execution.

**Figure 2 sensors-22-07750-f002:**
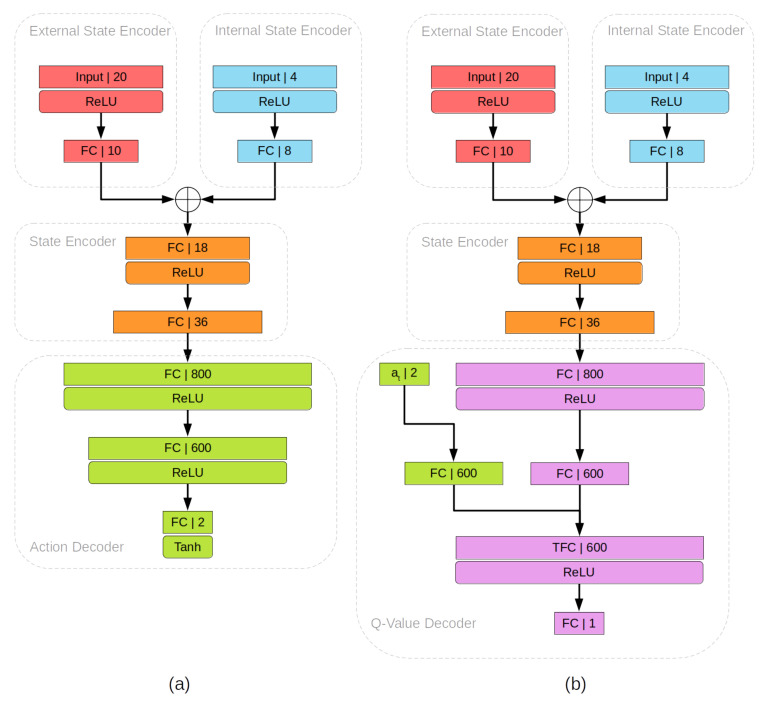
(**a**) Visualization of Actor Network architecture in TD3. (**b**) Visualization of Critic Network Architecture in TD3. FC represents a fully connected layer, numbers represent the output parameters of the respective layer, TFC represents the transformation fully connected layer, ⨁ represents concatenation.

**Figure 3 sensors-22-07750-f003:**
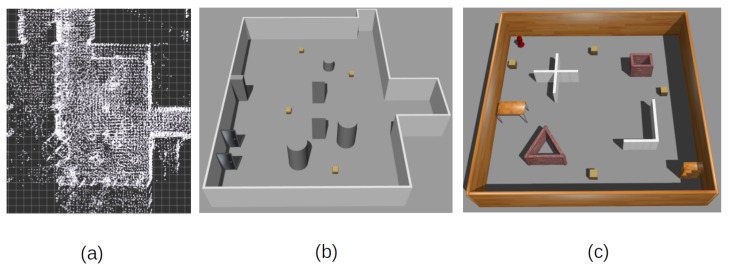
(**a**) Point cloud mapping of the cleaning environment in Riga Technical University. (**b**) ROS Gazebo simulated representation for expert demonstration collection. (**c**) ROS Gazebo simulated environment used for training of the neural network.

**Figure 4 sensors-22-07750-f004:**
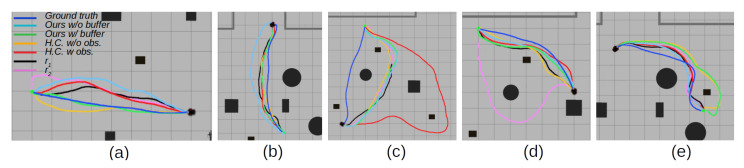
Examples of experimental results in a simulated environment with policy trajectory comparison to the recorded ground truth. The ground truth trajectories were used as samples in the proposed neural network training. (**a**) An example with no obstacles between the robot and a goal. (**b**) An example with a compromised start location. (**c**,**d**) Examples with obstacles between the robot and the goal. (**e**) An example where the learned policy does not follow the expert trajectory, but still exhibits the expert feature properties.

**Figure 5 sensors-22-07750-f005:**
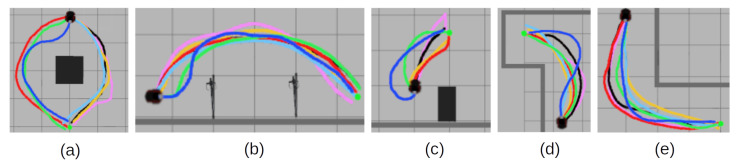
Examples of experimental results in a simulated environment with policy trajectory comparison to the recorded ground truth. The ground truth trajectories were not used as samples in the proposed neural network training. The handcrafted scenarios show the similarity to expert motion even in previously unseen scenarios. (**a**,**b**) Examples with obstacles between the robot and the goal. (**c**) An example with a compromised start location. (**d**,**e**) Examples of navigating around corners.

**Figure 6 sensors-22-07750-f006:**
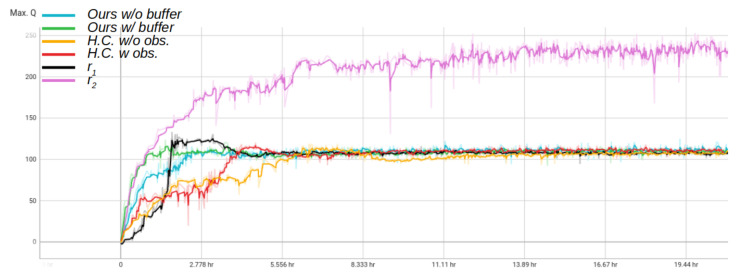
A training time representation of max. Q value per training for each policy. The training time was executed for approximately 20 h for each policy.

**Figure 7 sensors-22-07750-f007:**
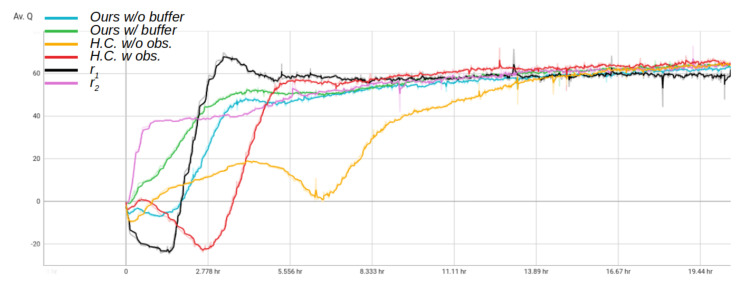
A training time representation of average Q value seen per each training for each policy. The training time was executed for approximately 20 h for each policy.

**Table 1 sensors-22-07750-t001:** Network parameters and structure of actor network.

	Actor Network	
**Layer**	**Input Size**	**Output Size**
External State Encoder:		
Linear	20	20
ReLU	20	20
Linear	20	10
Internal State Encoder:		
Linear	4	4
ReLU	4	4
Linear	4	8
State Encoder:		
Concatenation	10 + 8	18
Linear	18	18
ReLU	18	18
Linear	18	36
Action Decoder:		
Linear	36	800
ReLU	800	800
Linear	800	600
ReLU	600	600
Linear	600	2
Tanh	2	2

**Table 2 sensors-22-07750-t002:** Network parameters and structure of both critic networks.

	Critic Network	
**Layer**	**Input Size**	**Output Size**
External State Encoder:		
Linear	20	20
ReLU	20	20
Linear	20	10
Internal State Encoder:		
Linear	4	4
ReLU	4	4
Linear	4	8
State Encoder:		
Concatenation	10 + 8	18
Linear	18	18
ReLU	18	18
Linear	18	36
Q-Value Decoder:		
Linear	36	800
ReLU	800	800
$Linear_{state}$	800	600
$Linear_{action}$	2	600
TFC	600 + 600	600
ReLU	600	600
Linear	600	1

**Table 3 sensors-22-07750-t003:** Training parameter values.

Parameter	Value	Measurement Units
$r_{g}$	100	-
$r_{c}$	−100	-
$o_{v}$	0.6911	-
$o_{\omega }$	0.4445	-
$\phi_{v}$	1.1073	-
$\phi_{\omega }$	0.6615	-
$\eta_{\delta }$	0.3	m
$\eta_{l}$	0.3	m
$\lambda_{\Delta }$	1	degrees
*b*	20	-
$v_{max}$	1	m/s
$v_{min}$	0	m/s
$\omega_{max}$	1	r/s
$\omega_{min}$	−1	r/s
seed	0	-
max. episode length	500	steps
$H_{batch}$	40	samples
$E_{batch}$	2	samples

**Table 4 sensors-22-07750-t004:** Experiments in simulated environment.

	Ours w/o Buffer	Ours w/ Buffer	H.C. w/o Obs.	H.C. w/ Obs.	$r_{1}$	$r_{2}$
Mean $v_{\delta }$	0.17	0.20	0.13	0.12	0.02	−0.14
S.D. $v_{\delta }$	0.32	0.32	0.33	0.35	0.42	0.49
Mean $\omega_{\delta }$	0.29	0.23	0.18	0.19	0.37	0.43
S.D. $\omega_{\delta }$	0.36	0.36	0.36	0.36	0.35	0.34

**Table 5 sensors-22-07750-t005:** Experiments in simulated environment.

	Ours w/o Buffer	Ours w/ Buffer	H.C. w/o Obs.	H.C. w/ Obs.	$r_{1}$	$r_{2}$	Expert
Mean $step_{\delta }$	−2.34	**−4.04**	8.78	4.71	4.03	77.36	-
S.D. $step_{\delta }$	27.03	**26.69**	65.30	67.86	54.94	137.99	-
Mean $r_{ep}$	**67.96**	65.64	49.79	56.15	65.85	4.33	-
S.D. $r_{ep}$	**44.22**	51.41	149.26	155.51	80.91	224.71	-
Mean *v*	0.76	0.78	0.61	0.60	0.55	0.22	0.59
S.D. *v*	0.36	0.35	0.42	0.46	0.46	0.39	0.31
Mean |ω|	0.69	0.65	0.58	0.56	0.78	0.92	0.52
S.D. |ω|	0.34	0.35	0.37	0.37	0.33	0.22	0.32
Mean L2	35.60	**34.60**	48.92	34.96	39.45	64.38	-
S.D. L2	**47.06**	48.31	97.01	48.73	125.59	133.20	-
∑L2	5696.70	**5536.64**	7826.77	5593.84	6312.74	10,301.18	-

## Data Availability

Not applicable.

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
