# Peer review of "Leveraging Expert Demonstration Features for Deep Reinforcement Learning in Floor Cleaning Robot Navigation"

_sensors, 2022, doi:10.3390/s22207750_

Round 1

Reviewer 1 Report

This manuscript presents an approach based on Deep Reinforcement Learning (DRL) for the actual application of a mobile cleaning robot navigation system. The authors demonstrated their practice in a straightforward and detailed way—initially, they collected demonstrations of robot motion trajectories in a simulation environment in the cleaning robot domain. Based on the results, they extract the most relevant motion features about the distance to obstacles and the heading difference towards the navigation goal. Each feature weight is optimized for the collected data, and the obtained values are assumed to represent the expert navigation's optimal motion. They proposed a reward function based on the feature values to train a policy with semi-supervised DRL. An immediate reward is calculated based on the closeness to the expert navigation. Finally, the authors compare their results with others related to the literature and claim that the results show the viability of their approach regarding robot navigation and the reduced training time.

The authors did a good job. Their experiment is well described and will undoubtedly advance the state-of-the-art in this specific application of mobile devices guided by artificial intelligence. Despite that, I have some suggestions that may improve the final version of this manuscript.

·         I would like to see some comparisons of the main work described in the related works section. A description of the main desired features would help readers to understand this table.

·         An in-depth evaluation of the main similar proposals would enrich the present manuscript.

·         Finally, it would be nice to understand the improvements the authors propose for their approach to evasion and where their research will head to.

I enjoyed reading this manuscript and believe it will contribute to many other groups researching the same topic. 

Author Response

We would like to thank the reviewer for the time and effort spent to evaluate our proposed approach and the valuable comments on how to improve the manuscript. We have taken these comments into account, and hopefully improved the manuscript to facilitate all of them.

Q: I would like to see some comparisons of the main work described in the related works section. A description of the main desired features would help readers to understand this table.

A: We have specified the desired features of our approach in the introduction of related works, to make the selection of the related works more clear.

Q: An in-depth evaluation of the main similar proposals would enrich the present manuscript.

A: We have added additional experiments and their visualization in Fig4 and Fig5, to better present the learned behavior of combining the collected trajectories and random exploration in the training. This visualization directly compares the taken overall trajectories in the experiments over previously seen trajectories, as well as trajectories “out of distribution” with similar approaches.

Q: Finally, it would be nice to understand the improvements the authors propose for their approach to evasion and where their research will head to.

A: In Fig5, previously unseen scenarios are considered, with evasion around obstacles performed directly by the learned policy. This means that the policy is capable of learning generalizable evasion strategy. The future direction of the research is discussed in the Conclusions section paper. For additional evasion strategies with more dynamic obstacles, we plan to introduce LSTM or GRU based architecture to maintain the history information of the environment, motions in it as well as the motion of the robot. This has been specified in the Conclusions section.

Reviewer 2 Report

The experiment results mainly compare the training time of the proposed method with other different models.  Please add experiments which can illustrate the superiority of the proposed means in Floor Cleaning Robot Navigation.

Author Response

Q: The experiment results mainly compare the training time of the proposed method with other different models. Please add experiments which can illustrate the superiority of the proposed means in Floor Cleaning Robot Navigation.

A: We would like to thank the reviewer for the time and effort spent evaluating our proposed approach and the valuable comments on how to improve the manuscript. We have taken these comments into account and hopefully improved the manuscript to facilitate all of them. We have added additional experiments and their visualization in Fig4 and Fig5, to better present the learned behavior of combining the collected trajectories and random exploration in the training. This visualization directly compares the taken overall trajectories in the experiments over previously seen trajectories, as well as trajectories “out of distribution” with similar approaches. In Fig5, previously unseen scenarios are considered, with evasion around obstacles performed directly by the learned policy. This means that the policy is capable of learning a generalizable local motion strategy based on a strategy designed explicitly for floor cleaning features. The experiments in Section 6.1 also show quantifiable results of the learned motion policy and its capability in replicating the motions of expert experience, while being able to adapt to new situations. This allows the creation of a full navigation stack for robot control in floor cleaning scenarios in the future, where the local navigation layer is replaced by a neural network with a motion policy designed for floor cleaning tasks.

Reviewer 3 Report

The manuscript gives a compressive demonstration of a Deep Reinforcement Learning (DRL) based robot control and navigation system.  The training of the presented neural network using two comparative approaches combined with reward methods derived from expert demonstrations.

The manuscript has a clear structure, a well-explained problem presentation and a sophisticated introduction to the state of the art and the mathematical description of system parameters and attributes.

Chapter 2 gives an overview about related work about robot navigation and expert features extraction. Chapter 3 addresses the reward mechanisms and chapter 4 details the architecture of the neural network in more detail, while chapter 5 gives an explanation network training with the involved parameters. Finally, chapter 6 gives an overview of the experimental results and chapter 7 the conclusion of the related work and future work.

The manuscript presents a compressive research summary for deep reinforcement learning and requires from the reader a deep knowledge in neural network technologies.

The manuscript provides an important topic of neural network-controlled robot systems. In general, the manuscript can be approved for publication without a full evaluation from my side of the theoretical mathematical formulations.

References are checked only randomly. All selected references found.

In my opinion, the manuscript can be released for publication.

Author Response

We would like to thank the reviewer for the time and effort spent to evaluate our proposed approach and the valuable comments on how to improve the manuscript. We have taken these comments into account, and hopefully improved the manuscript to facilitate all of them.

Reviewer 4 Report

Leveraging Expert Demonstration Features for Deep Reinforcement Learning in Floor Cleaning Robot Navigation
> However, BC has reported issues when dealing with generalization[3].
> Therefore, we propose to implement the idea, borrowed from BC, to use the expert demonstrations and assume them as a guideline for training a neural network for cleaning robot navigation by using DRL.
BC can be used when the application is "simple", so an error committed by the agent does not lead to severe consequences.
Actually, deep learning based BC is already widely researched.
The following is simply one example.
https://towardsdatascience.com/using-deep-learning-to-clone-driving-behavior-51f4c9593a57
(1)
So, compared with previous approaches, I am wondering what is novel about the authors' approach.
The novel point is that the BC based approach is used in Floor Cleaning Robot Navigation?
Because the main idea behind Behavioral Cloning is to learn a policy for an MDP given expert demonstrations, and that is exactly what the authors are trying to do.
> To counter this issue, a Twin Delayed Deep Deterministic Policy Gradient (TD3) architecture uses two critic networks instead of one and chooses the values of the critic that provides the lower values, thus limiting the overestimation [18].
(2)
DDPG is that bad? And also Twin Delayed Deep Deterministic Policy Gradient (TD3) really solve the problem of overestimating the Q-value?
> In our implementation, we propose to combine the BC-based approach of evaluating the performance against the designed expert motion features in the cleaning robot domain, which are expressed as a reward function, thus allowing to use DRL for learning a motion policy in unknown environments without the need to learn the reward function through IRL.
I am afraid it is simply a reinforcement learning with "expert motion features as a reward function".
You just use "designed expert motion features" as a reward function.
(3)
Also, I suspect about the effectiveness of the results because it is performed in simulated environments.
The simulated environment should be similar to "Floor Cleaning Robot Navigation".
It is unclear what simulated environments are used for evaluation.
The authors have to describe their simulated environments used for the experiments more clearly.

(4)
Some suggestions:
As for English, more sentences are in a passive voice.
In the past, many researchers used passive voice, but nowadays, we usally use active voice in the academic paper.
Also, some titles like below are widely used in the papers by non-native speakers, but actually they are not clear to native speakers.
Related Works --> Related Work
http://ngrams.blogspot.com/2012/04/scientific-publications-related-work-vs.html

Experimental Results --> Results
https://www.englishforums.com/English/ExperimentResultExperimental-Result/bdzcld/post.htm

Author Response

Q: BC can be used when the application is "simple", so an error committed by the agent does not lead to severe consequences.
Actually, deep learning based BC is already widely researched.
The following is simply one example.
https://towardsdatascience.com/using-deep-learning-to-clone-driving-behavior-51f4c9593a57
(1)
So, compared with previous approaches, I am wondering what is novel about the authors' approach.
The novel point is that the BC based approach is used in Floor Cleaning Robot Navigation?
Because the main idea behind Behavioral Cloning is to learn a policy for an MDP given expert demonstrations, and that is exactly what the authors are trying to do.

A: The main approach is the introduction of BC approach in DRL through expert demonstrations in replay buffer. As the reviewer has pointed out, BC is well researched and capable of achieving good results in “simple” tasks, but generalizability is an issue when tasked with situations in a new environment. Therefore, DRL is used in our approach to explore the environment and learn from it, based on a reward function, obtained from the expert demonstration, while still using these demonstrations in DRL. Therefore, we are not performing BC per se, but rather using DRL with an evaluation similar to that of BC.
Q:  To counter this issue, a Twin Delayed Deep Deterministic Policy Gradient (TD3) architecture uses two critic networks instead of one and chooses the values of the critic that provides the lower values, thus limiting the overestimation [18].
(2)
DDPG is that bad? And also Twin Delayed Deep Deterministic Policy Gradient (TD3) really solve the problem of overestimating the Q-value?

A: In the authors' experience and also as reported in the relevant papers, the DDPG approach is rather unstable. While the optimality of the obtained policy would be the same as with the TD3 approach, often the overestimation of Q values leads the optimization process of DDPG to local optimum early on in the training process. This requires very careful tuning of all DDPG parameters to output consistent performance of DDPG. By “dampening” the Q value in TD3, the optimization is given more time to explore the surrounding policy and stabilize the learning of the policy. Therefore, the TD3 approach is the preferred way of learning a motion policy in this setting, with the downside that another critic network is required, which makes the neural network larger. However, when deploying the motion policy, only the actor-network is required, meaning that DDPG and TD3 methods will have the same amount of parameters.
Q: In our implementation, we propose to combine the BC-based approach of evaluating the performance against the designed expert motion features in the cleaning robot domain, which are expressed as a reward function, thus allowing to use DRL for learning a motion policy in unknown environments without the need to learn the reward function through IRL.
I am afraid it is simply a reinforcement learning with "expert motion features as a reward function".
You just use "designed expert motion features" as a reward function.

A: In the relevant related work, the reward function is generally designed by hand, with explicit decisions of the desired behavior. Then, a reward is collected by moving a certain way. A set behavior is rewarded either positively or negatively. In our approach, we first design the explicit features we want our motion policy to focus on. Then we propose, how to calculate these features dynamically, based on expert motion data and not by designing them by hand. After this, the reward is not conditioned on the behavior as it is done in similar approaches, but rather a similarity to the feature. That means, that we do not reward, for instance, the speed value of the robot directly, but rather the difference of the speed to the one calculated by the floor cleaning robot motion feature (as is often done in BC settings). Therefore, in our approach, we do not use a designed expert motion reward function, but rather a “similarity score” to the expert motion feature as a reward function.

Q: Also, I suspect about the effectiveness of the results because it is performed in simulated environments.
The simulated environment should be similar to "Floor Cleaning Robot Navigation".
It is unclear what simulated environments are used for evaluation.
The authors have to describe their simulated environments used for the experiments more clearly.

A: We have specified the simulation environments used in the experiments and added additional experiment visualizations to hopefully make the distinction of the simulation setting clear. We have added additional experiments and their visualization in Fig4 and Fig5, to better present the learned behavior of combining the collected trajectories and random exploration in the training. This visualization directly compares the taken overall trajectories in the experiments over previously seen trajectories as well as trajectories “out of distribution” with similar approaches.

Q: Some suggestions:
As for English, more sentences are in a passive voice.
In the past, many researchers used passive voice, but nowadays, we usally use active voice in the academic paper.
Also, some titles like below are widely used in the papers by non-native speakers, but actually they are not clear to native speakers.
Related Works --> Related Work
http://ngrams.blogspot.com/2012/04/scientific-publications-related-work-vs.html
Experimental Results --> Results
https://www.englishforums.com/English/ExperimentResultExperimental-Result/bdzcld/post.htm

A: We have taken the suggestions into account and updated the paper accordingly.

Round 2

Reviewer 2 Report

 Thanks for authors' answer and the additional experiments can present the better behavior of the proposed method. 

Reviewer 4 Report

I don't have any other issues.